# Diet and gut microbiome enterotype are associated at the population level in African buffalo

Claire E. Couch[1][✉], Keaton Stagaman [2], Robert S. Spaan [3], Henri J. Combrink[1], Thomas J. Sharpton[2,4,6], Brianna R. Beechler [5,6] & Anna E. Jolles [1,5,6]

Studies in humans and laboratory animals link stable gut microbiome "enterotypes" with long-term diet and host health. Understanding how this paradigm manifests in wild herbivores could provide a mechanistic explanation of the relationships between microbiome dynamics, changes in dietary resources, and outcomes for host health. We identify two putative enterotypes in the African buffalo gut microbiome. The enterotype prevalent under resource-abundant dietary regimes, regardless of environmental conditions, has high richness, low between- and within-host beta diversity, and enrichment of genus *Ruminococcaceae-UCG-005*. The second enterotype, prevalent under restricted dietary conditions, has reduced richness, elevated beta diversity, and enrichment of genus *Solibacillus*. Population-level gamma diversity is maintained during resource restriction by increased beta diversity between individuals, suggesting a mechanism for population-level microbiome resilience. We identify three pathogens associated with microbiome variation depending on host diet, indicating that nutritional background may impact microbiome-pathogen dynamics. Overall, this study reveals diet-driven enterotype plasticity, illustrates ecological processes that maintain microbiome diversity, and identifies potential associations between diet, enterotype, and disease.

[1] Department of Integrative Biology, Oregon State University, Corvallis, OR, USA. [2] Department of Microbiology, Oregon State University, Corvallis, OR, USA. [3] Department of Fisheries & Wildlife, Oregon State University, Corvallis, OR, USA. [4] Department of Statistics, Oregon State University, Corvallis, OR, USA. [5] Carlson College of Veterinary Medicine, Oregon State University, Corvallis, OR, USA. [6] These authors jointly supervised this work: Thomas J. Sharpton, Brianna R. Beechler, Anna E. Jolles. [✉]email: claire.couch@oregonstate.edu

In the past two decades, rapid advances in high-throughput sequencing technologies and improvements in computational systems have facilitated a radical transformation of our understanding of the rich microbial ecosystems that exist in symbiosis with mammalian hosts[1–5]. Extensive studies in humans and laboratory animals have demonstrated widespread links between the gut microbiome, nutrition, and infectious disease[6–9], and the methodologies advanced in these studies now afford us the opportunity to traverse the frontier of wildlife microbiomes[5,10]. Studying the microbiomes of wild species has the potential to clarify the ecological and evolutionary drivers of host–microbiome dynamics by contextualizing these relationships in natural ecosystems. To our knowledge, all mammals host complex communities of microbial life in and on their bodies, and many of these bacterial relationships provide essential host functions. The importance of microbial symbioses may be amplified in ruminants, as they tend to harbor highly abundant and diverse gut microbiota[11,12] and rely heavily on microbes to extract energy from indigestible plant matter[13]. Although the gastrointestinal microbiomes of domestic ruminants have been studied extensively in the effort to maximize agricultural productivity[14,15], the ecological and evolutionary significance of the relationships between wild ruminants and their commensal microbiota are less understood. Understanding the structure and dynamics of microbial communities in wild ruminant populations has the potential to highlight ecological patterns in host–microbiome relationships, and to strengthen theoretical understanding of how these relationships respond under changing environmental conditions.

Some studies in humans and other primates suggest that gut communities can be classified into one of several stable "enterotypes", defined by the relative abundance of key bacterial taxa, which are shaped by long-term diet and resilient to short-term perturbation[16–18]. However, the frequency and time scale at which dietary change can cause individuals to transition between enterotypes is unclear, and some dispute the validity of the enterotype paradigm altogether[19,20]. In frugivorous great apes, shifts between enterotypes coincide with seasonal resource availability[21], raising questions regarding the relative importance of seasonal environmental change compared with seasonal dietary shifts. The enterotype paradigm has not been thoroughly explored outside of omnivorous and frugivorous primates (i.e. humans and great apes), therefore the degree of seasonal stratification and plasticity in the gut microbiomes of other mammalian taxonomic and dietary groups, including wild herbivores, remains unclear. Recent work suggests that geographic and seasonal variation in the gut microbiomes of several wild herbivore species could be driven at least in part by dietary shifts[22–25], but it is difficult to separate seasonal environmental changes from the associated seasonal changes in diet. Understanding the drivers of enterotype dynamics in wild herbivores, and ultimately the nutritional and physiological impacts of enterotypes, could be of great ecological significance. Herbivores provide a functional link between primary producers and predators and can profoundly shape the landscapes they inhabit[26]. Additionally, many herbivores are highly dependent on microorganisms to digest and synthesize nutrients[27], therefore linking potential microbiome enterotypes with nutrition and physiology in these species could provide important insights regarding the resilience of herbivores and their ecosystems to environmental changes.

In humans, it is unclear whether host enterotypes respond differentially to disease[19,28]. However, human gut microbiome structure has been associated with enteric infections[29], and with pathogen invasion across multiple body sites including the respiratory tract[30,31], thus clarifying enterotype–disease relationships could contribute greatly to predictive and therapeutic health measures. The prevalence of well-studied infectious diseases[32] and complex communities of gut symbionts[33] that characterize many wild ruminant populations imply that these populations could offer excellent opportunities to study differential associations between host enterotypes and infectious diseases. Moreover, understanding the relationships between pathogenic and commensal microbes could have relevant applications to conservation and epidemiology[33], in addition to contributing to our basic understanding of disease ecology.

In this work, we explore microbiome stability and structure within and between individuals over time in a population of African buffalo (*Syncerus caffer*), a long-lived social ruminant. We analyze longitudinal fecal microbiome data collected from 72 buffaloes between February 2014–February 2017 during which they were exposed to natural and artificial fluctuations in resource availability, parasites, and pathogens. Our investigation is framed around three central hypotheses:

(i) Dietary regime is the dominant driver of microbiome structure in wild African buffalo. Resource-abundant dietary regimes allow for greater microbial richness, similar to the relationship between net primary productivity and animal species richness[34].

(ii) Microbiomes are resilient to diet regime shifts. Resilience may manifest at the individual level (i.e. individualized resilience to resource changes) or at the population level (i.e. predictable population-level shifts in dominant enterotype). Individualized resilience could be driven by physiological host factors that select for or against certain bacterial taxa depending on dietary conditions, whereas population-level resilience could be driven by diet-driven microbial extinction within hosts followed by recolonization facilitated by microbial dispersal between hosts.

(iii) The relationships between commensal microbes and pathogens depend on dietary regime. Diet-driven changes to the microbiome associate with variation in the magnitude and direction of microbiome–pathogen associations, similar to environmentally mediated changes to interspecific interactions observed in macroscopic ecosystems. Changes to microbiome–pathogen relationships could be mediated by resource-driven variation in the immune system, similar to environmentally driven changes observed in predator–prey dynamics[35]. Additionally, changes in host diet could alter competition for resources between commensal and pathogenic microbes or enable facilitative interactions[36].

## Results

**Alpha diversity and composition.** A total of 44,103 unique amplicon sequence variants (ASVs) were identified to the genus level from 426 samples. We removed 36,412 ASVs that were not classified to the genus level, as we were primarily interested in analyzing known bacterial taxa. Of these unclassified ASVs, 10,512 (29%) occurred only once in the entire dataset, and 33,092 (91%) occurred <10 times in the entire dataset. Clustering results were robust to the removal of these ASVs, likely because they were rare (median relative abundance 2.34e−7, maximum 1.6%). After filtering, rarefied samples contained a median of 731 ASVs (minimum 88, maximum 1905, standard deviation 385.1). The median number of bacterial genera identified in each sample was 90 (minimum 30, maximum 154, standard deviation 20.7), and the top 10 most abundant bacterial genera across all samples were *Solibacillus*, *Ruminococcaceae UCG-005*, *Lysinibacillus*, *Ruminococcaceae UCG-010*, *Bacillus*, *Romboutsia*, *Rikenellaceae RC9 gut group*, *Christensenellaceae R-7 group*, *Bacteroides*, and *Ruminococcaceae UCG-013*.

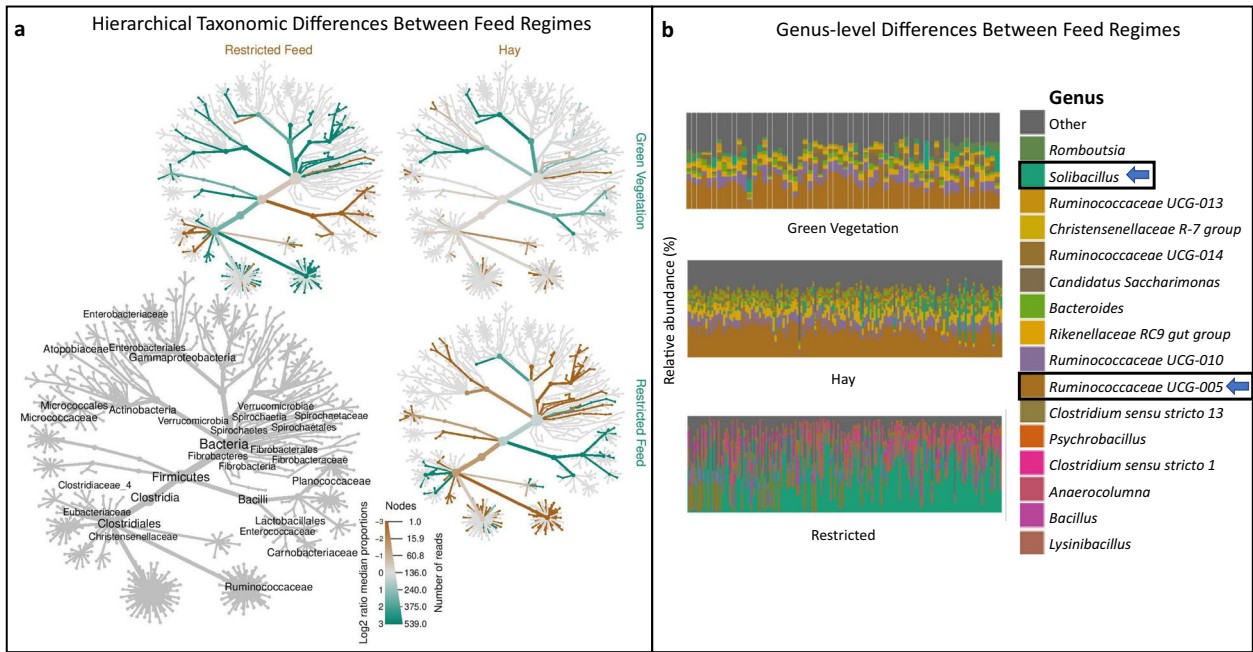

**Fig. 1 Diet regime drives structural variation in the African buffalo gut microbiome. a** Annotated taxonomic hierarchies show relative enrichment of taxa from the kingdom through family levels across pairwise comparisons between diets. The hierarchies with colored nodes represent pairwise comparisons between the dietary regimes listed on the x-axis and y-axis. Lineages that are highlighted in brown or green indicate log 2-fold increase in median abundance of that lineage in the x-axis group or the y-axis group, respectively. The bottom left hierarchy provides a key with labels for all taxa that significantly differed in at least one of the pairwise comparisons (false discovery rate-adjusted Wilcoxon rank sum $q < 0.05$), with size of labels corresponding to number of genera contained in that node. **b** Relative abundance of the most common genera across individuals in each diet regime. Under the restricted regime, 72% of samples were dominated by genus *Solibacillus*. In contrast, *Ruminococcaceae UCG-005* was the most abundant genus in 88% of the samples from the hay regime, and 90% of the samples from the green vegetation regime.

**Dietary regime is the primary driver of microbiome structure in African buffalo**. PAM clustering and Calinski–Harabasz index comparisons demonstrated an optimal number of 2 clusters that were 85% correlated with dietary regime. Within-individual repeatability of cluster membership was estimated to be 0 ($p = 0.50$), but generalized linear-mixed model (GLMM) results demonstrated a significant relationship between cluster membership and diet regime at the population level ($p < 2e−16$). PAM clustering results were robust to the removal of unclassified genera, with 98% of samples falling into the same cluster regardless of whether all ASVs or only known genera were used. Variance partitioning showed that diet alone explained 48% of the variation in microbiome composition, as opposed to 3% explained by individual ID alone and 4% explained by capture period. We observed a clear shift in dominant taxa between the restricted regime compared with the hay and green vegetation regimes (Fig. 1a). The most common bacterial genus under the hay and green vegetation regimes was *Ruminococcaceae-UCG-005*, which was the most abundant genus in 90% of samples from the green vegetation group and 88% of the hay group. In contrast, under the restricted regime, the majority (72%) of samples were dominated by genus *Solibacillus* (Fig. 1b). Due to the similarity observed at the genus level between the green vegetation and hay-fed regimes, downstream linear discriminate analysis of effect size (LEfSe) was used to identify higher level taxonomic differences that distinguished these two groups from the restricted regime. LEfSe and visualization with metacoder showed that higher level taxonomic trends (i.e. phylum-family levels) aligned with the genus-level distinction (Fig. 1a, Supplementary Fig. S1). We also confirmed that when LEfSe was run at the class-phylum levels, it was robust to filtering ASVs not classified to the genus level. Eight out of the nine phyla that were identified as differentially abundant between restricted vs. hay/green vegetation regimes in the

filtered dataset were also differentially abundant in the unfiltered dataset.

**Opposite shifts in alpha and beta diversity across diet regimes**. While gamma diversity (population-level richness) did not change significantly across dietary regimes (Kruskal–Wallace $p = 0.076$), we observed significant opposing shifts in alpha diversity (individual-level richness) and beta diversity (inter-individual and intra-individual compositional differences) between the restricted regime versus the hay and green vegetation regimes (Fig. 2). GLMMs showed that alpha diversity was reduced in the restricted vegetation regime compared with the green vegetation ($p = 0.00136$) and hay regimes ($p = 7.12e−7$) but did not differ between the hay and green vegetation regimes ($p = 0.354$). In contrast, permutation tests for homogeneity of multivariate dispersions showed that beta diversity (Bray–Curtis distance to mean) was elevated during the restricted vegetation regime compared with the green vegetation regime (permuted $p$-value $= 0.002$) or hay (permuted $p$-value $= 0.001$) but did not differ significantly between the hay and green vegetation regimes (permuted $p$-value $= 0.056$).

**Diet-driven enterotypes**. We defined two host enterotypes based on the alpha and beta diversity dynamics observed at the population level over shifts in feeding regime: a restricted nutrition enterotype, defined by dominance of *Solibacillus*, low alpha diversity and high beta diversity, and a high nutrition enterotype, defined by *Ruminococcaceae UCG-005* dominance, high alpha diversity, and low beta diversity, prevalent across the hay and green vegetation regimes. According to redundancy analysis (RDA) results, microbiome composition associates with both individual animal identity ($p = 0.001$) and diet ($p = 0.001$), but variance partitioning demonstrated that diet explained a much

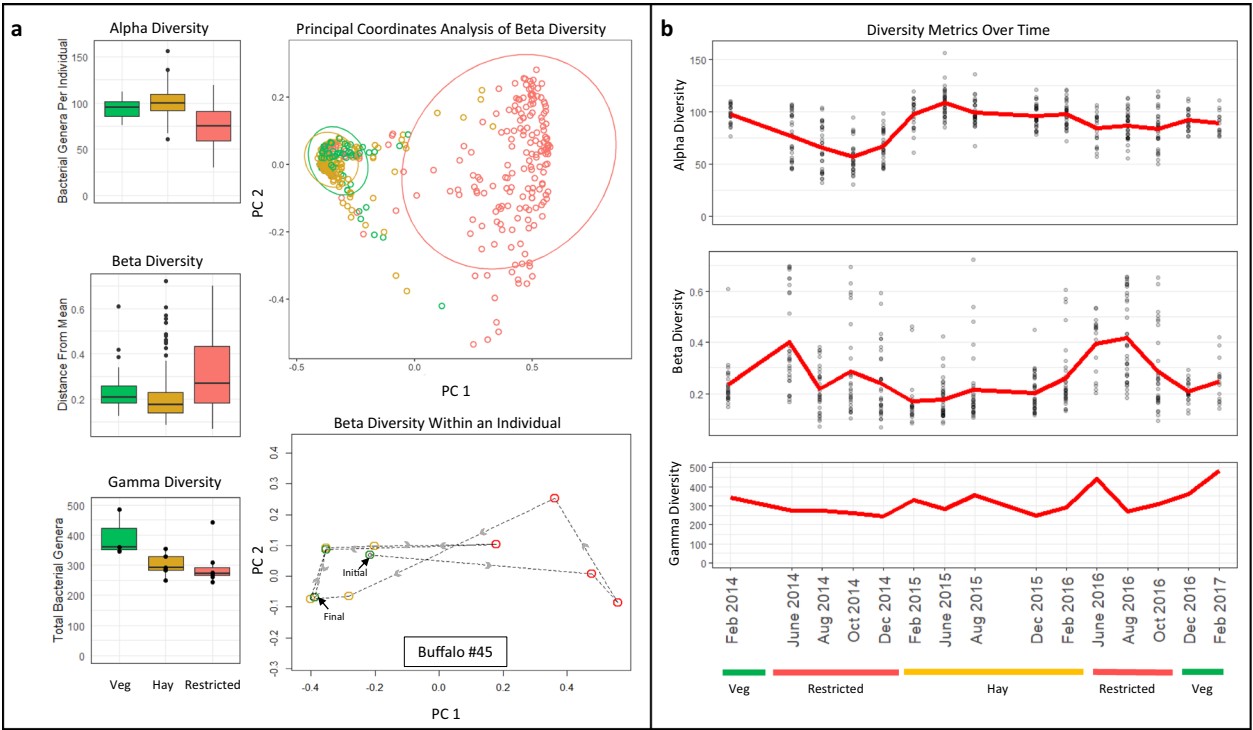

**Fig. 2 Microbiome alpha and beta diversity showed opposing patterns of variation across dietary regimes. a** Diversity metrics in panel **a** are colored by feed regime (green indicating green vegetation, yellow indicating hay, and red indicating restricted feed). For each box plot, the lower and upper hinges correspond to the first and third quartiles. The upper and lower whiskers extend from the hinges to the largest and smallest values no more than 1.5 IQR from the hinge. Outliers beyond these points are plotted individually. Gamma diversity did not significantly change across the three dietary regimes (Kruskal–Wallace $p = 0.076$, $n = 15$ time points), whereas alpha diversity was significantly reduced in the restricted regime compared with hay ($p = 7.12e-7$) and green vegetation ($p = 0.00136$) regimes (generalized linear mixed model, $n = 426$ samples). Beta diversity for each time point was greater in the restricted regime than either the hay (permuted $p$-value $= 0.001$) or green vegetation (permuted $p$-value $= 0.002$) regimes (permutation test for homogeneity of multivariate dispersion, $n = 426$ samples). A single individual with near-complete longitudinal sampling (Buffalo #45) was selected to demonstrate temporal shifts at the individual level, with time points connected chronologically by dashed lines. Individual shifts in microbiome composition aligned with population-level shifts in dietary availability. **b** Temporal patterns in alpha and beta diversity appeared to oppose each other, while gamma diversity remained stable over time. Source data are provided as a Source Data file.

**Table 1 Covariates that significantly associated with microbiome composition based on the envfit variable selection analysis and the final CCA model.**

| Covariate Group | Significant in envfit analysis of all samples | Significant in envfit analysis of restricted nutrition samples | Significant in envfit analysis of hay/green vegetation samples | Retained and significant in final CCA |
|---|---|---|---|---|
| Disease | BVDV | MB, BVDV, strongyle burden | BHV | MB, BHV, BVDV |
| Nutrition | Hct, MG | None | MG | None |

larger proportion of variation in the data than individual identity (48% versus 3%). Moreover, canonical correspondence analysis (CCA) results showed that individual effects disappeared when other host factors (age, sex, infection status) were accounted for, therefore we attribute the significant effect of individual identity to physiological variation rather than as evidence of persistent individual microbiome signatures.

**Associations with disease**. We identified three respiratory infections that associated with microbiome variation after accounting for capture period, diet, age, sex, and nutritional variables (Table 1). Preliminary analysis with *envfit* suggested that microbiome beta diversity associates with *M. bovis* incidence, bovine viral diarrhea virus (BVDV), and strongyle burden during the restricted dietary regime, and with bovine herpes virus (BHV) during the green vegetation regime. The subset of host traits and diseases that were selected via *envfit* analysis were included in an

initial CCA (Eq. (2)), which was then subject to bidirectional selection resulting in Eq. (1).

$$\text{CCA}\big(\text{genus table} \sim \text{MB} + \text{BVDV} + \text{BHV} + \text{Condition}(\text{Age}) + \text{Condition}(\text{Sex}) + \text{Condition}(\text{Capture Number}) + \text{Condition}(\text{Diet})$$

$$(1)$$

Of the constraining variables, only MB, BHV, and BVDV were retained in the final model, and all of these variables were significant based on a permutation test (Table 1). Results from the preliminary envfit analysis suggested that microbiome composition associated differentially with BHV, BVDV, and *M. bovis* depending on diet, and this was supported by visualizing principal coordinate analysis of disease associations within each dietary regime (Fig. 3).

In addition to multivariate community-level shifts associated with disease, we also identified specific bacterial genera associated with disease incidence or presence within the dietary regimes in

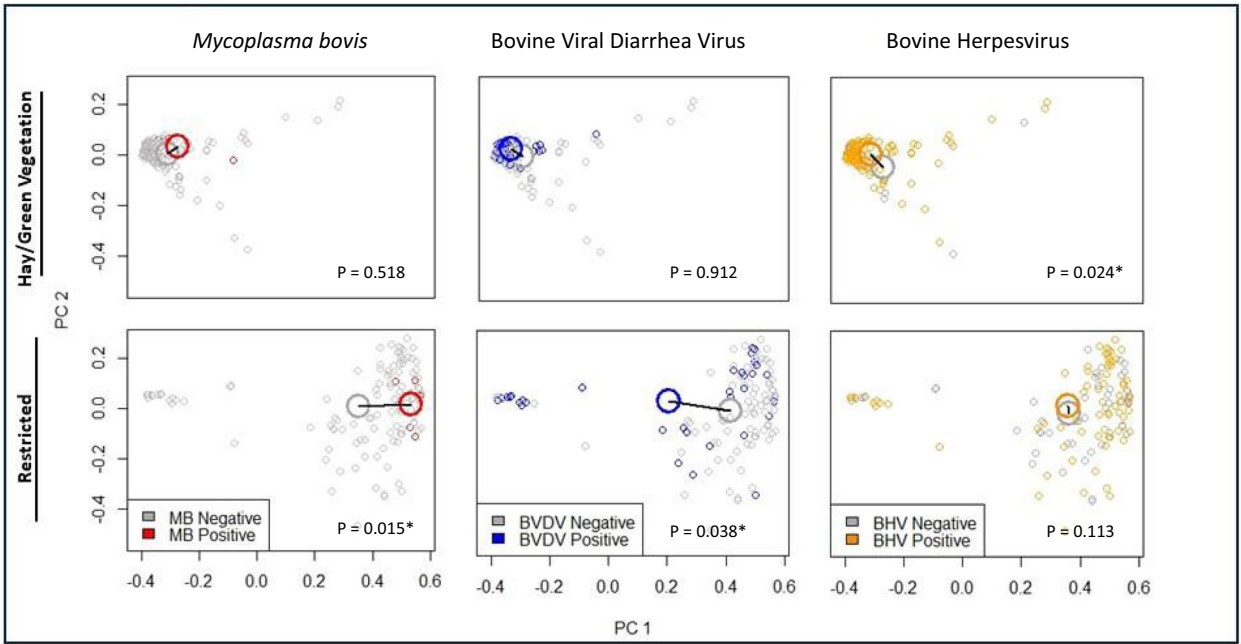

**Fig. 3 Three respiratory infections associated with gut microbiome composition in African buffalo based on canonical correspondence analysis:** ***Mycoplasma bovis* (MB), bovine viral diarrhea virus (BVDV), and bovine herpesvirus (BHV).** Differences in the magnitude and/or direction of association between disease and the microbiome were observed between the restricted regime and the hay/green vegetation regime based on visualization and envfit analysis of principal coordinate analysis. This plot shows differences in microbiome community composition associated with infection status based on principal coordinate analysis of Bray–Curtis distances for the restricted and hay/green vegetation regimes. Analysis by envfit was based on 134 samples in the restricted regime (5 MB positive, 35 BVDV positive, 85 BHV positive) and 177 samples in the hay/green vegetation regime (10 MB positive, 48 BVDV positive, 159 BHV positive) for which complete disease datasets were available. *p*-values for envfit analysis of infections within each group are shown on each panel. Colored points indicate seroconversion (*M. bovis*, positive = red) or presence (BVDV, positive = blue; BHV, positive = yellow), and gray points indicate lack of seroconversion (*M. bovis*) or absence (BVDV and BHV). Small points indicate individual samples, and larger circles indicate the means for infected and uninfected groups. Source data are provided as a Source Data file.

which envfit analysis suggested correlation with community variation. GLMM results identified one genus associated with *M. bovis*, seven with BHV, and six with BVDV (Table 2).

## Discussion

This study reveals plasticity in the African buffalo gut microbiome that is closely tied to changes in dietary resources. We demonstrate striking fluctuations in microbiome community structure between resource-restricted and resource-rich diet regimes (i.e. dry season conditions with no supplementary feeding versus high availability of green vegetation or supplementary hay), and very little difference between resource-rich diets (green vegetation versus supplementary hay). Uniquely, this study demonstrates that abundant dietary intake associates with predictable outcomes for microbiome community structure at the host population level regardless of the dietary source (i.e. natural vegetation or supplementary feed) and regardless of conditions (i.e. dry season or wet season). Shifts between dietary regimes had by far the most profound influence on microbiome structure compared to host traits, individual ID, and infectious disease. These findings contribute to the understanding of diet-driven enterotype formation of the gut microbiome. During resource abundant dietary regimes, individual hosts harbor similar, species-rich microbiomes. Individuals lose microbial taxa during periods of resource restriction, resulting in reduced alpha diversity, but microbial gamma diversity is maintained at the host population level via increased beta diversity among hosts. Enterotype shifts are predictable at the population level based on diet regime. However, within each diet regime, microbiome variation is largely stochastic with respect to individual. Taken

together, our findings on diet-driven shifts in alpha and beta diversity suggest that within-host extinction of microbial taxa is driven by resource restriction, but that recolonization via microbial sharing between hosts facilitates microbiome resilience once abundant dietary conditions return.

Loss of microbial richness during dietary restriction does not appear completely stochastic with respect to microbial taxa. Rather, our findings suggest that certain taxa are adapted to resource-restricted regimes and are more likely to survive in the African buffalo gut. During periods of enriched resource availability (hay or green vegetation), gut communities were characterized by high taxonomic richness (Fig. 2a), low beta-diversity (Fig. 2b), and enrichment of genus *Ruminococcaceae UCG-005* (Fig. 1). This genus is widespread in the gut microbiomes of wild ruminants[25,37–39], and has been linked to environmental/dietary heterogeneity in several of these studies[38,39]. During restricted resource periods, gut communities showed relatively low taxonomic richness and increase in relative abundance of *Solibacillus*. In goats and cattle, genus *Solibacillus* has also been associated with reduction in forage intake, possibly as an adaptive response to increased dietary variability[40,41]. We did not explicitly measure behavioral changes in this study, but free-ranging buffalo exhibit seasonal changes in behavior and herd cohesion[42]. It is possible that if such behavioral patterns are present in our study herd, they could mediate changes in microbial transmission dynamics, potentially explaining some of the diet-associated patterns in richness and composition found in this study[43].

In addition to reduced richness and predictable increases in certain taxa, our findings show increased composition variability between individuals during periods of restricted nutrition, suggesting reduced stability under restricted dietary conditions.

**Table 2 Bacterial genera associated with respiratory infections within each dietary regime based on GLMMs.**

| Disease | Diet regime | Number of negative and positive samples | Number of genera evaluated | Positively associated genera | Negatively associated genera |
|---|---|---|---|---|---|
| Mycoplasma bovis | Restricted | 7 positive 155 negative | 373 | Viridibacillus (est. = 2.74, q = 0.0062) | None |
| Bovine herpesvirus | Hay/Green Vegetation | 193 positive 26 negative | 468 | Saccharofermentans (est = 0.74, q = 0.013), Clostridioides (est = 35.9, q = 0.013) Catenisphaera (est = 14.6, q = 0.037) Patulibacter (est = 21.30, q = 0.045) q = 0.032) | Domibacillus (est = −2.62, q = 0.013) Paucisalibacillus (est = −2.312, q = 0.045) Pseudoflavonifractor (est = −1.68, q = 0.032) |
| Bovine viral diarrhea virus | Restricted | 41 positive 160 negative | 391 | None | Actinobacter (est = −0.53, q = 0.046) Phascolarctobacterium (est = −1.37, q = 0.040) Psychrobacillus (est = −0.66, q = 0.040) Methanocorpusculum (est = −24.4, q = 0.042) DTU089 (est = −4.29, q = 0.017) Ureibacillus (est = −0.74, q = 0.049) |

Alongside predictable increases in taxa adapted to resource restriction (e.g. *Solibacillus*), restriction may drive community destabilization resulting in random loss of taxa and greater microbiome dissimilarity between individual hosts. A possible mechanism could be that reduction in the taxa that typically dominate under resource-rich conditions (e.g. *Ruminococcaceae UCG-005*) destabilizes gut communities, leading to stochastic variation. Another potential mechanism of divergence could be differential resource selection among hosts during periods of dietary restriction, leading to greater variability of selective pressures on microbiome communities among individuals. Whether the mechanism driving increased compositional variability during resource restriction is stochastic or selective, the increased variability between individuals may provide refugia for diverse microbes, enabling rapid recolonization and return to the rich, homogenous microbiomes typical during resource abundant periods regimes via transmission between hosts.

The associations we observed between diet, microbiome, and disease in this study highlight the importance of considering associations between microbiome diversity and disease within the context of broader spatio-temporal resource distribution. Our findings suggest that the magnitude and direction of associations between microbiome variation and disease depends on the dietary context. Our analysis showed that *M. bovis*, BHV, and BVDV associate with microbiome composition, but that the magnitudes and directions of those associations differ between resource-abundant (hay or green vegetation) versus resource-restricted dietary regimes (Fig. 3). Preliminary analysis with envfit detected microbiome associations with *M. bovis* and BVDV during resource restricted but not resource-rich time periods, whereas BHV associated with microbiome composition during resource-rich periods. These infections are known to have immunosuppressive effects in cattle, but may also result from immunosuppression[44–46], therefore it is possible that the apparent relationships with the gut microbiome are mediated via changes to the host immune system. We also identified bacterial taxa putatively associated with incidence or presence of infections. This finding suggests that non-pathogenic taxa may indicate susceptibility to or infection by multiple pathogens and could therefore be considered as ecological indicator species for the gut and a potential surveillance tool. However, we recommend interpreting the associations between individual taxa and disease with caution due to the manifold environmental and host variables not accounted for in this study. Future studies should focus on clarifying causal directionality of the pathogen–microbiome relationships identified in this study, and elucidating interactions between pathogen communities and commensal microbiomes in the context of host nutrition.

The study of wildlife microbiome communities has the potential to contribute to conservation efforts in addition to expanding our understanding of host–microbe ecology. We demonstrate resource-driven enterotype plasticity in the gut microbiomes of a social herbivore population, underscoring the dynamic interplay between environmental change and microbiome communities. This study demonstrates how microbial diversity can be maintained in a host population over the long term, despite resource-driven reductions in individual-level alpha diversity. We found that the increase in beta diversity among individuals during dietary restriction maintains population-level gamma diversity. This suggests that a large, well-connected host population could be important for recolonizing and restoring individual-level alpha diversity following dietary restriction. These results imply that host population size and connectivity may be important for maintaining microbiome diversity within a population, especially as climate change and habitat loss drive more dramatic fluctuations in resource availability. Our findings could inform future efforts to utilize microbiome information as a form of noninvasive monitoring to guide conservation efforts for wildlife populations threatened by environmental change. Additionally, understanding

commensal–pathogen dynamics in natural host populations under a range of environmental conditions is key to bridging the gap between laboratory studies and observational surveys of wildlife microbiomes. As demonstrated in our results, altered gut enterotypes and differential relationships between pathogens and commensal microbiome communities can manifest under fluctuating resource regimes. Considering current trajectories of human-caused environmental change, understanding how resource availability shapes wildlife gut microbiome communities, and clarifying the downstream consequences for host health and disease, will be crucial aspects of efforts to manage wildlife populations. Moreover, broadening our understanding of long-term microbiome community dynamics in natural host populations can help contextualize our understanding of microbiome dynamics in humans and domestic animals.

## Methods

**Field methods**. Our study was located in Kruger National Park (KNP), a 19,000 km² reserve located in northeastern South Africa with ~30,000 free-ranging African buffalo. For this study, we longitudinally sampled a herd of buffalo contained in a 900-ha enclosure near Satara rest camp (Fig. 4). This enclosure was designed to exclude large predators (e.g. lions, leopards), rhinoceros, and elephants, but contained other herbivores and small mammalian predators typical of the ecosystem, in addition to the buffalo. The approximate size of the herd at any given time was between 50 and 65 buffalo depending on births and deaths. Every 2–4 months over the course of 3 years (February 2014–February 2017), each individual was captured for physical examination and biological sampling, resulting in a total of 15 captures during which 72 buffalo were sampled, 62 of which were sampled more than once. This period overlapped with three different dietary regimes: periods of high green vegetation driven by seasonal rainfall, restricted forage availability driven by seasonally low rainfall and vegetation, and supplementary feeding during periods of extended drought (Table 4). Due to an unusual period of drought that extended beyond the dry season of 2014 through October of 2016, grass inside the enclosure was limited. Dry season versus wet season conditions were identified based on normalized difference vegetation index (NDVI) values. We used 16-day composite, 250-m resolution NDVI data from MODIS for the North American Carbon Program (MODIS for NACP, https://accweb.gsfc.nasa.gov/). NDVI data was extracted to the 900-ha enclosure (Fig. 1) and mean NDVI for each capture period was calculated statistic using R packages raster[47], sp[48], rgdal[49], maptools[50], and rgeos[51]. To prevent mortalities, buffalo were provided with supplemental feed. From February 2015 through February 2016, the herd was provided with 800 kg bale of lucerne, five 370 kg bales of grass hay and 20 kg of concentrated game pellets (Alzu feeds) per day to retain body condition typical under wet season conditions. Following this period, feed was reduced to mimic dry season conditions by removing the concentrate and gradually reducing the alfalfa to 2/3 bale until the rains began in December 2016. For simplicity, the abundant supplemental feed period is referred to throughout this work as "hay" and dry season periods with reduced or no supplemental feed are referred to as "restricted". During captures, each animal was sedated with a high potency opioid (Thianil or etorphine hydrocholoride) and azaperone at dosages appropriate to weight and sex[52].

Blood was collected by jugular venipuncture for hematocrit measurements[53], biochemical analysis, and disease assays[54,55]. Age in years was calculated from incisor emergence and tooth wear[56], and sex was determined visually. Body condition score (BCS) was assessed by palpation of four regions (ribs, hips, spine, and base of tail), each of which were scored on a scale of 1–5 and then averaged across the four regions. This method has been shown to correlate with kidney fat index and total hematocrit[57]. Rectal samples of feces were collected directly from sedated animals using sterile gloves, and samples were immediately placed on ice for 5–8 h during travel to the laboratory, where they were aliquoted for gastrointestinal parasite counts[57], fecal chlorophyll analysis[58], and microbiome analysis, then froze at −80 °C. Blood was collected via jugular venipuncture directly into vacutainer tubes with (plasma, whole blood) or without (serum) heparin, and stored on ice for transport. Blood was centrifuged at 5000×g for 10 min, and plasma, and serum were pipetted off the cellular layer and frozen at −80 °C until they were used to run serum biochemistry panels[54] and test for a suite of respiratory pathogens[55] (Table 3). *Mycobacterium bovis*, the causative agent of bovine tuberculosis, was evaluated using the BOVIGAM test according to manufacturer's instructions (ThermoFisher Scientific product no. 63326). Seroconversion of adenovirus (AD3), parainfluenza virus (PI3), BHV, and *Mycoplasma bovis* were tested using the Bio-X IPAMM sandwich ELISA kit. Bovine diarrhea virus (BVDV) and bovine respiratory syncytial virus (BRSV) were tested using the Bio-X BVDV and Bio-X BRSV ELISA kits. Samples were considered positive for pathogen antibodies if antibody titers exceeded threshold absorbance values calculated using the quality control procedures outlined in each kit. Incidence was calculated as a binomial variable for acute infections (AD3, PI3, *M. bovis*) and was assigned a 1 if an animal seroconverted from the previous capture period and 0 if the animal had not seroconverted[45]. All animal work for this study was approved by the institutional animal care and use committee at

Oregon State University, ACUP project number 4478, and by KNP, ACUP project number JOLAE1157-12.

**Microbiome sample processing and sequencing**. Fecal samples were collected within 30 min of sedation from captured animals, placed on ice within 15–30 min of collection, and frozen at −80 °C within 8 h of collection. Genomic DNA was extracted using the DNEasy PowerSoil kit following manufacturer's instructions, with the addition of a 10 min incubation step at 65 °C immediately prior to bead-beating for 20 min. We amplified a 450 bp region of the V3/V4 region of the bacterial 16S gene. Extracted DNA was subject to a first round 16S PCR amplification using the following primers: 16S Forward Primer 5′-TCGTCGGCAGCGTC AGATGTGTATAAGAGA CAGCCTACGGGNGGCWGCAG-3′ and 16S Reverse Primer 5′-GTCTCGTGGGCTCGGAGATGTGTA TAAGAGACAGGACTACH VGGGTATCTAATCC-3′. PCR reactions were amplified with GoTaq Hot Start Polymerase (Promega, Madison, WI) following manufacturer's suggested use. PCR cycling conditions were as follows: an initial melt of 94 °C for 3 min followed by 35 cycles of amplification with a 94 °C for 30 s, 55 °C annealing step for 1 min, and a 68 °C extension step for 1.5 min. A final, 5-min extension step was included following the last cycle. Amplicons were cleaned, indexed, and normalized by the Oregon State University Center for Genome Research and Biocomputing prior to sequencing on two runs of the Illumina Miseq V2 platform using Miseq control software v2.6.2.1, resulting in 250 bp paired-end reads. We employed the DADA2 workflow implemented in the dada2 R package (version 1.12.1)[59] to identify ASVs, trim adapter sequences, and remove chimeras. Raw sequence data were processed through the dada2 pipeline using the following trimming parameters: trimLeft = c (17, 21), truncLen = c(250,250), maxN = 0, maxEE = 2, truncQ = 2. Default parameters were used for estimating error parameters using learnErrors, and chimeras were removed using removeBimeraDenova (method = "consensus"). Prior to statistical analysis, data from the two runs were combined, each sample was randomly subsampled to a depth of 20,000 reads, and samples with sequencing depth below this cutoff were excluded from downstream analysis[60]. ASVs not identified to the genus level were also removed from downstream analysis. Relative abundance of these unclassified ASVs were rare, with a median relative abundance of 2.34e−07, and none exceeding 2% of reads.

**Statistical analysis**. All statistical analyses and visualizations were conducted in R (version 3.6.1)[61] unless otherwise specified. After subsampling to an even depth, ASVs were merged by genus, resulting in 543 unique genera. The function estimate_richness in the phyloseq package (version 1.30.0)[62] was used to calculate genus-level richness. GLMMs assuming Poisson distribution were used to perform pairwise comparisons of genus richness among the three nutritional regimes while controlling for capture number using the lme4 package (version 1.1.21)[63]. Bray–Curtis distance calculations of genus abundance tables, principal coordinate analysis, and initial data visualization were performed in phyloseq. Sequencing runs were compared using PERMANOVA stratified by capture number. Because sequencing run had no effect on composition (R squared = 0.08, p = 0.52) after accounting for capture number, it was not included as a variable in downstream analysis. To identify a priori clusters in the microbiome data[16], we used the partition around medoids (PAM) algorithm[64] on Bray–Curtis distances using the pam function in the cluster package[65] and then identified the optimum number of clusters by calculating the Calinski–Harabasz index for 2–20 clusters[66]. PAM clustering was repeated including all rarefied ASVs including those not classified to the genus level to ensure that clustering was robust to the ASV filtering. Repeatability of enterotype membership across captures within individuals was analyzed using the GLMM-based function rptBinary with a logit link in the rptR package[67]. The relationship between enterotype and diet was estimated using a GLMM model with cluster membership as the response variable, abundant vs. restricted diet as a fixed response variable, and individual ID as a random effect. Differences in the most abundant bacterial genera for each of the three dietary regimes were visualized using the plot_bar function in phyloseq. We assessed the significance of dietary regime and individual ID by running distance-based RDA using the dbrda function in vegan (version 2.5.6)[68], while controlling for capture number using the Condition() argument. Proportion of variance explained by diet, individual ID, and capture period were calculated and visualized using the vegan function varpart. The vegan function betadisper was used to compute multivariate dispersion (average distance to median) for each group, and these results were then compared using the permutest function. LEfSe was used to identify diet-associated differences at higher taxonomic levels (i.e. phylum, class)[69]. Based on observed patterns of composition and alpha diversity, we used "high nutrition" (hay and green vegetation) versus restricted as the main grouping variables, and finer-scale diet regime (hay, green vegetation, and restricted) as the subgrouping variables in the LEfSe algorithm. Individual effects were ignored in the LEfSe algorithm. Differences in taxonomic abundances between the three feed regimes were visualized using the heat_tree_matrix function in the metacoder package[70]. LEfSe was also conducted using all rarefied ASVs including those not classified to the genus level to verify that phylum-level results were robust to the ASV filtering.

Our initial dataset included a large number of host covariates (Table 3, Figs. S2, S3, S4), so we followed the variable selection method used by Flannery and Stagaman[71] to identify potential covariates of interest. The goal of this approach is to reduce a large covariate dataset to a manageable number of covariates and thus

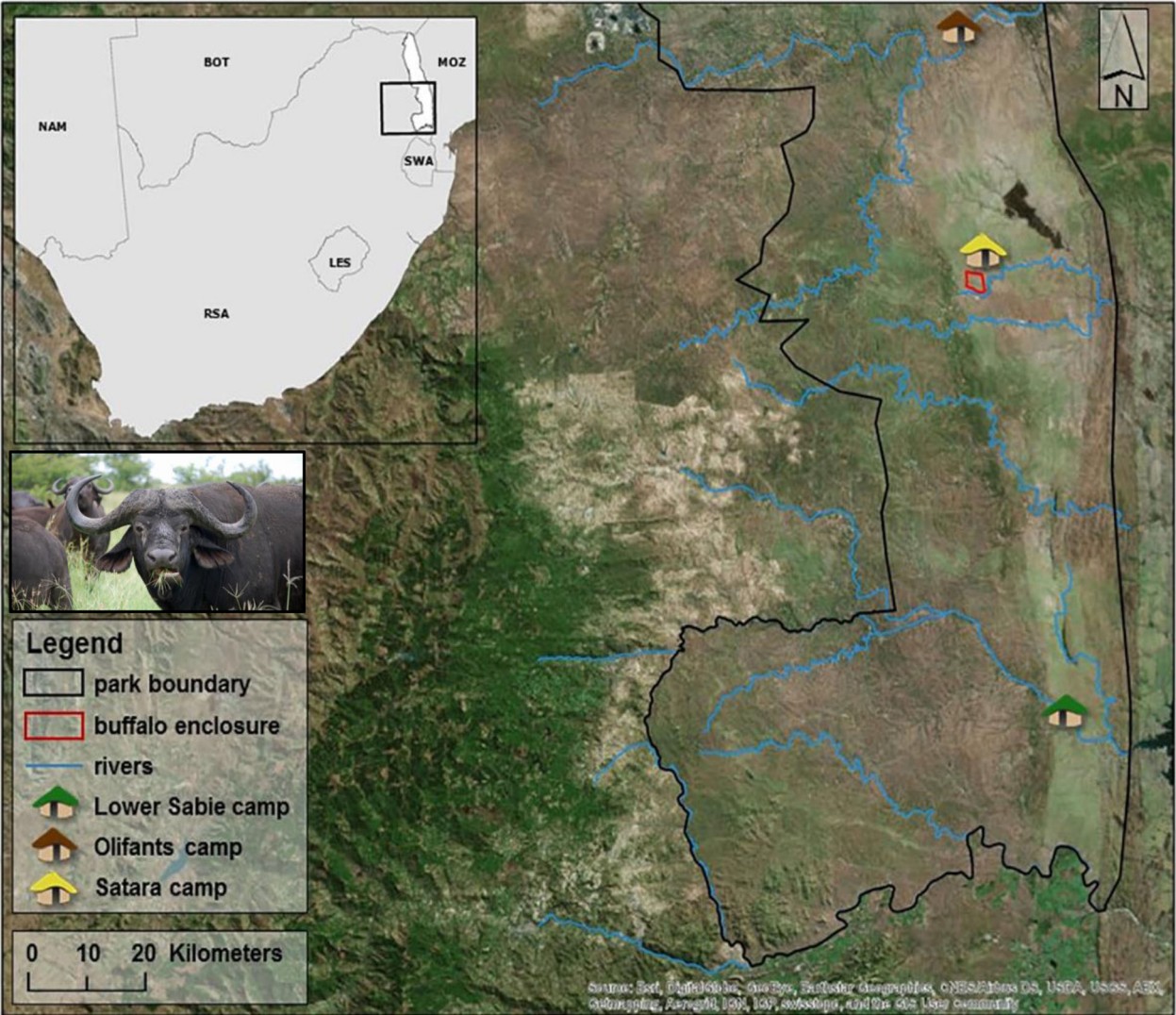

**Fig. 4 The African buffalo population in this study was located near Satara camp in Kruger National Park, South Africa.** Buffalo resided in a 900 ha (3.5 square mile) double-fenced enclosure designed to exclude large predators. The population was originally isolated in 2001 as part of a disease-free breeding program within the park. The insets show the location of Kruger National Park within South Africa (top left) and an African buffalo from the park (photo by Robert Spaan). Buffalo were either darted from a helicopter or corralled in the northeast corner of the enclosure prior to sedation and sampling. The background map was created using ArcGIS® software by Esri. ArcGIS® and ArcMap™ are the intellectual property of Esri and are used herein under license. Copyright © Esri. All rights reserved. For more information about Esri® software, please visit www.esri.com.

| Variable category | Included in *envfit* analyses | Included in CCA | Included in CCA with diet interaction term |
|---|---|---|---|
| **Table 3 Host covariates included in the variable selection process and retained for downstream CCA analysis.** | | | |
| Disease | Bovine tuberculosis (TB) status, Parainfluenza-3 incidence, Adenovirus-3 incidence, *Mycoplasma bovis* (MB) incidence, bovine respiratory syncytial virus (BRSV) incidence, bovine herpesvirus (BHV) status, bovine viral diarrhea virus (BVDV) status, strongyle burden (log transformed), coccidia burden (log transformed), trichuris burden (log transformed) | | MB incidence, strongyle burden, BHV status, BVDV status |
| Nutrition | Body condition score (BCS), fecal chlorophyll, hematocrit (Hct), calcium, phosphorous, magnesium (MG), total protein | Hct | MG |

We performed envfit analysis on PCA objects generated from Bray–Curtis distance matrices within the restricted dietary regime, the hay/green vegetation regimes, and pooled samples. For each variable group, host covariates were considered "significant" within a given dataset if envfit analysis yielded $p < 0.05$. Permutations within the envfit algorithm were stratified by capture number to account for temporal effects. Chronic respiratory disease status (tuberculosis, bovine herpesvirus, bovine diarrhea virus) was coded as 0 if negative and 1 if positive. Acute respiratory infection incidence (parainfluenza-3, adenovirus-3, *Mycoplasma bovis*, bovine respiratory syncytial virus) were coded as 0 if they did not seroconvert and 1 if they did seroconvert between captures. Gastrointestinal parasite burdens were measured in eggs per gram of feces for each individual.

**Table 4 Distribution of sample sizes and sampling dates from across the three diet regimes.**

| Capture number | Month/Year | Supplementary feed | Mean NDVI | Diet regime | Number of fecal samples |
|---|---|---|---|---|---|
| 1 | February 2014 | None | 0.64 | Green vegetation | 25 |
| 2 | June 2014 | None | 0.40 | Restricted | 30 |
| 3 | August 2014 | None | 0.29 | Restricted | 30 |
| 4 | October 2014 | None | 0.28 | Restricted | 28 |
| 5 | December 2014 | Started introducing feed | 0.25 | Restricted | 31 |
| 6 | February 2015 | Fed | 0.47 | Hay | 22 |
| 7 | June 2015 | Fed | 0.32 | Hay | 37 |
| 8 | August 2015 | Fed | 0.27 | Hay | 31 |
| 9 | December 2015 | Fed | 0.25 | Hay | 37 |
| 10 | February 2016 | Fed | 0.25 | Hay | 32 |
| 11 | June 2016 | Reduced feed | 0.25 | Restricted | 20 |
| 12 | August 2016 | Reduced feed | 0.24 | Restricted | 36 |
| 13 | October 2016 | Reduced feed | 0.23 | Restricted | 30 |
| 14 | December 2016 | None | 0.21 | Green vegetation (224.6 mm rainfall, not reflected in NDVI) | 22 |
| 15 | February 2017 | None | 0.70 | Green vegetation | 15 |

Each individual was sampled only once at each capture period. All samples were used for alpha and beta diversity analysis and envfit analysis. Only samples that were accompanied by complete covariate datasets were included in the CCA. Normalized difference vegetation index (NDVI) was summarized as mean NDVI values for the MODIS pixels overlapping the boma during the time period most closely preceding the capture. Dietary regime was classified as "restricted" if NDVI was <0.5 and little or no supplementary feed was provided, with the exception of December 2016, which had exceptionally high rainfall and abundant new vegetation growth that was not reflected in NDVI measurements. Dietary regime was classified as "hay" if abundant supplementary feed was provided (see the "Methods" section) and as green vegetation otherwise.

avoid overfitting of explanatory models or identification of spurious correlations given the large number of covariates relative to the number of available samples[71]. Because our dataset included many incomplete observations, reducing the number of covariates (and thus the number of potentially incomplete data fields) also allowed us to retain a larger sample size for analysis. The number of covariates was reduced by first using unconstrained ordination (principal coordinate analysis) and vector fitting to identify covariates that correlated with principal components. Based on composition and alpha diversity results, we expected the principal coordinate axes to differ between the high nutrition and low nutrition regimes, therefore we performed principal coordinate analyses separately for the high nutrition and low nutrition groups as well as the pooled dataset, followed by vector fitting on each of the three ordinations (high nutrition, low nutrition, and pooled). For each of the three ordinations, envfit analyses were run separately for host disease and nutrition covariates on samples for which complete cases were available using the strata argument to control for capture number (Table 3, Fig. 4). Based on the envfit results (Table 1), a model was constructed for constrained correspondence analysis (CCA) using the vegan cca function (Eq. (2)). If envfit analysis demonstrated correlation between a covariate and either of the first two PCA axes in the pooled data or either diet regime, that covariate was included in the global CCA model. Dietary regime (high versus low nutrition) was included as an interaction term for covariates that were significant in the envfit analysis of the pooled data but did not meet the significance threshold in one of the dietary regimes, or that were significant in one of the dietary regimes but not in the pooled dataset (Tables 1 and 3). Individual ID was also included as a fixed effect in the initial model. Due to missing disease incidence and hematocrit data, we included only data from captures 3–13 for the CCA, resulting in 258 samples collected from 53 unique individuals. Bidirectional step selection was conducted using the vegan ordistep function to identify a final CCA model that controlled for age, sex, capture number, and diet as conditional variables. Significance of the final model parameters were assessed by performing a permutation test using anova.cca(by = "terms")

$$
\begin{aligned}
\text{CCA}(\text{genus table} \sim\ & \text{Individual ID} + \text{Hct} + \text{MG*Diet} + \text{BVDV*Diet} + \text{MB*Diet} + \text{BHV*Diet} \\
& + \text{Strongyles*Diet} + \text{Condition(Age)} + \text{Condition(Sex)} + \text{Condition(Capture Number)} \\
& + \text{Condition(Diet)}
\end{aligned}
\tag{2}
$$

After analyzing the relationships between disease and diet covariates and overall microbiome composition using ordination-based approaches, we identified the specific bacterial genera associated with these relationships using Tweedie compound Poisson generalized linear-mixed models (GLMMs) implemented in the cpglmm function of the cplm package[72,73]. For each genus, we compared GLMMs of the general structure of Eq. (3) with a reduced model (4) using the anova function to determine the significance of association between each genus and covariate of interest. GLMMs were run only within the dietary regime for which the envfit analysis identified significant correlation with the covariate of interest. Because GLMMs were run separately for each covariate of interest, sample sizes were slightly larger than for the envfit analysis (Table 2). Model results were ignored taxa with insufficient prevalence or variation for model convergence. We ran three sets of GLMMs assessing genus correlations with (1) *M. bovis* seroconversion in the restricted regime, (2) BVDV presence in the restricted regime, and (3) BHV presence in the enriched (hay/green vegetation) regimes.

$$
\text{Genus}_i \sim \text{Disease} + (1|\text{Age}) + (1|\text{Sex}) + (1|\text{Capture}|\text{Number}) \tag{3}
$$

$$
\text{Genus}_i \sim 1 + (1|\text{Age}) + (1|\text{Sex}) + (1|\text{Capture Number}) \tag{4}
$$

**Reporting summary**. Further information on research design is available in the Nature Research Reporting Summary linked to this article.

## Data availability
Microbiome sequence data are available in the NCBI SRA database (BioProject ID PRJNA694651, https://www.ncbi.nlm.nih.gov/bioproject/PRJNA694651/). MODIS for NACP data is available at https://accweb.gsfc.nasa.gov/. Source data are provided with this paper. All other datasets generated during and/or analyzed during the current study are available from the corresponding author on reasonable request. Source data are provided with this paper.

## Code availability
The analysis code that supports the findings of this study is available upon request from the corresponding author.

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

## Acknowledgements

This work was made possible by the assistance of Veterinary Wildlife Services at KNP, especially Peter Buss and Markus Hofmeyer. Hannah Tavalire, Caroline Glidden, Brian Dugovich, Danielle Sisson, Henri Combrink, Katherine Forssman, and countless others were instrumental in data collection. Microbiome 16S sequencing was conducted by the Center for Genome Research and Biocomputing at Oregon State University. Funding for microbiome sequencing was provided by the NSF GRFP (grant # 1840998), The American Genetic Association, and the American Association of Zoological Veterinarians (grant # 57). Buffalo disease and physiological data collection was funded by USDA-NIFA AFRI (grant # 2013-67015-21291) and by the UK Biotechnology and Biological Sciences Research Council (grant # BB/L011085/1) as part of the joint USDA-NSF-NIH-BBSRC Ecology and Evolution of Infectious Diseases program. C.E.C. was supported through the NSF GRFP (grant # 1840998), the ARCS Foundation Scholars Program (Oregon chapter), and the Robert and Clarice MacVicar Animal Health Award. K.S. and T.J.S, were supported through NSF grant #1557192.

## Author contributions

C.E.C. conducted initial PCR, statistical analysis, prepared the manuscript, and generated data figures. K.S. advised statistical analysis and data visualization and provided manuscript review and editing. R.S.S. processed and analyzed NDVI data and generated Fig. 1. H.J.C. conducted fecal chlorophyll analysis. T.J.S. supervised data analysis and provided manuscript review and editing. B.R.B. and A.E.J. provided funding, project administration, conceptualization, and manuscript review and editing.

## Competing interests

The authors declare no competing interests.
