## [Peer Review File · Nature Communications]

Reviewers' comments:

Reviewer #1 (Remarks to the Author):

This study examines factors associated with the composition of the gut microbiota in wild African buffalo. Results include the identification of 'enterotype' clusters in the data associated with host diet, as well as associations between gut microbiota composition and respiratory pathogens. It is rare to see a study in which so many interesting data types (microbiome, diet, pathogen status, etc.) are analyzed, and the associations identified will be of broad general interest to the field. The manuscript is well-written and most of the analyses are well explained. However, there is one major issue regarding the identification of enterotypes, as it appears the authors have not tested for the emergence of microbiome clusters from the beta diversity results (revisions/additions suggested below). This issue should be addressed before publication.

In previous analyses, enterotypes have been identified in the data by clustering analyses agnostic to metadata variables. For example, the original Arumugam et al paper use Partitioning Around Medoids (PAM) clustering to identify the enterotypes. See: <https://enterotype.embl.de/enterotypes.html> for a detailed explanation of the methods and R code. Although the data presented here in the PC plots strongly suggest that the samples do cluster into 2 groups, the PC axes alone (which do not explain all the variation in the underlying beta-diversity measures) are not sufficient to demonstrate this. I would recommend conducting the analyses outlined in the original Enterotype paper (link above). For example, it would be interesting to see what the optimal number of clusters is using Calinski-Harabasz indices. Because the enterotype categories fundamentally rely on the statistical methods employed, it would be best to replicate previous analyses as closely as possible. Then, it would be interesting to test whether clusters that emerge from the data without respect to metadata variables (e.g., diet) are associated with the metadata variables. I think this approach would be a more robust way of addressing the questions 1) do clusters exist in the data and 2) if yes, are the clusters associated with diet/infection status. The analyses already present in the paper could still be presented, as it is also of interest to test for associations between microbiota composition and metadata variables directly.

Fecal sampling: Please provide more information on fecal sampling. 8 hrs is a long window for samples to be sitting at ambient conditions before freezing during which time the composition of the microbial community is quickly changing. How did this time vary among samples? Is this variation associated with recovered composition? If this information is not available, this could be discussed as a potential caveat.

Figure 1: If available, it would be interesting to include sample locations as points/stars/etc on the map (or as n='some number' for each sampling site), so that the spatial distribution of samples can be readily assessed by readers.

Figure 2: It would be interesting to assess taxonomic/genus-level differences between enterotypes as identified by the Arumugam et al. analyses suggested above, in addition to the differences between feed regimes shown here.

Figure 3: Similarly, it would be interesting to assess whether individuals shift between enterotypes (as identified by the Arumugam et al. analyses suggested above), and whether these shifts (if they exist) correspond to diet shifts.

Figure 4: It was difficult to assess significance of the difference between positive and negative for each panel. Would it be possible to include stats in the figure and/or legend? As noted for Figures 2 and 3, it would be interesting here to test whether infection status is associated with enterotypes (as identified by the Arumugam et al. analyses suggested above).

Reviewer #2 (Remarks to the Author):

Couch et al. have analyzed fecal microbiome composition collected over two years in African buffalo fed with supplemental diets. The framework of the study is interesting, the longitudinal data valuable, and the collected disease parameters useful. I don't have major criticisms of the paper, except a few things for the authors to consider (see below). However after reading through this mostly descriptive paper, I feel like I've not really learned anything particularly new about African buffalo gut microbiomes? For example, the study goes into detail showing that there are alpha and beta diversity differences between diet types, which is in itself not terribly surprising, but it doesn't describe clearly in what way they are different besides alpha and beta diversity measures? (see below) And how this is related to seasonal differences besides diet? In addition, I feel like the authors have a highly valuable dataset with collected disease parameters and body condition, but this section of the paper is unfortunately short and does not clearly outline what happens in the gut microbiome in individuals with disease.

Major comments

I hope the authors are aware of the controversy surrounding the term Enterotypes. The initial paper describing enterotypes in human gut microbiomes has been extremely criticized, to the point where it is often used as a cautionary example of "how to not analyze microbiome data". I have personally no problem with using this word, and I believe it is up to the authors to decide how to best describe their results. But thinking how to best help the authors with their revision, it may be worth to at least consider if they still feel like this word is the best way to present the differences they find.

How precisely does the microbiome differ with the diet regimes? Which taxa are associated with which diet, and do the results agree with previous diet studies? Why does one of the diet types lead to lower diversity? How does this relate to seasonal changes? The authors have conducted Lefse analyses but do not really present these clearly, except refer to Figure 2a which is not explained particularly well. Is this presented somewhere else and I missed it?

A very interesting aspect of this paper is the association tests with diseases. I feel like this section could

be stronger, however. What I've gathered from the presented results in figure 4 is that three diseases are associated with the gut microbiome. However, the authors could try to more clearly explain in what way the gut microbiome is different? Are there specific taxa associated with/without disease? What does it mean? Does it vary over the season or between the sexes? What about body condition and disease?

Minor comments

L109: The words "each animal" has been accidentally repeated.

L157 & L160: All ASVs not identified to genus level were removed and the rest merged within genera. I'm curious as to the reasoning behind this approach? I have not encountered it previously. Do the authors not miss out on a large proportion of the microbiome data if they require such precise taxonomic assignment? Seeing how the current databases, Silva, GreenGenes, etc are populated with bacteria obtained primarily from model organisms such as humans and mice, a large proportion of unculturable bacteria from wildlife are likely not present with genus-level-specific information. Because of this concern, I'm interested to hear about the reasoning to only keep ASVs with genus information in a wild mammal.

Related to this, a genus-specific approach also complicates the interpretation of the bar plots in Figure 2b. Do these proportions look very different when retaining the excluded genera?

L176: PairwiseAdonis package is not cited. In addition, this package seems to be under development still? The developer writes: "This is still a developing version -- results using interactions may not be right. Please validate." Are there any evidence and tests available that this function runs correctly as it says it might not be?

There's a large focus in the paper how the microbiome is associated with diet. However, diet differences are here directly associated with seasonal differences. Therefore, I'm curious how much seasonal changes play a role in the gut microbiome in addition to diet? For example there could be a large bacterial fluctuation present in the environment depending on whether it is dry or wet seasons, irrespective of diet. Several previous papers have found large associations between gut microbiome and season in wild mammals.

Figure 2: I'm not able to tell the differences between the brown, orange, and yellow colors in the bar plots. Consider using more distinct colors or fewer taxonomic groups in the legend. In addition, why only present 10 randomly selected samples from each feeding regime? I think all host individuals could be useful to present.

L367: The authors write that "These results imply that host population size and connectivity may be important for maintaining microbiome diversity within a population". My question is in what way have the authors analyzed associations with population size? I cannot find this approach in the paper.

Reviewer #3 (Remarks to the Author):

I appreciate the opportunity to read the manuscript submitted to Nature Communications by Couch et al. entitled, 'Diet drives gut microbiome enterotype shifts at the population level in wild African buffalo' and offer the following review:

In this study, the authors describe a two-year study, with sampling every 2-3 months, to assess the impact of seasonal fluctuations in diet on gut microbiome composition, pathogen occurrence, and parasite burden in a population of wild African buffalo inhabiting Kruger National Park in South Africa. The sampling frequency (17 capture periods), sampling size (50-65 buffalo per capture period), and numerous types of samples collected and analyzed represent a considerable research effort.

General comments

My main critique is that I'm unsure if this research study, as it's currently presented, significantly advances our understanding of how dietary shifts influence microbiome composition in wild mammals and/or how pathogen/parasite infections influence or are influenced by microbiome composition. As the authors mention in their manuscript, it's well established in the microbiome literature that humans and other primates exhibit gut microbiome plasticity in response to seasonality in diet. How does this study contribute novel ecological insight, other than demonstrating seasonality in gut microbiome composition for wild buffalos? Secondly, though the authors found two gut microbial "enterotypes" representative of resource rich and resource poor dietary conditions, it's unclear if there was a formal statistical analysis to identify enterotypes or if microbiome samples collected during the hay/green vegetation or restricted feed periods were de facto assigned to separate enterotypes.

Abstract

General comments: I recommend including more details about the study population (i.e., X individuals were longitudinally sampled from a long-term study population in Kruger National Park), sampling regime (e.g., 17 captures over 2 years), and the types of samples collected. For example, though the authors mention respiratory pathogens at the end of the abstract, the abstract does not explicitly mention that multiple types of samples were collected, in addition to fecal samples. Can you explicitly mention that resource rich and resource restricted periods associated with wet and dry seasons, respectively?

Line 5: I recommend rewording as "changes in dietary resources"

Lines 8 and 10: It's unclear if beta diversity refers to both intra-host, inter-host, or both inter- and intra-host microbial dissimilarities.

Lines 13-16: Was pathogen detection associated with resource rich periods or restricted dietary

conditions? Could respiratory pathogen detection be confounded by environmental conditions and not necessarily be associated with gut microbiome composition?

Introduction

Line 71: I recommend including much more detail about the buffalo study population and sampling regime.

Line 84: It's unclear to me how respiratory pathogen infection would be directly influenced by changes in gut microbiome composition.

Methods

Line 96: Include the total number of capture periods here (according to Table 1, there are 17). How many individual animals were longitudinally sampled, for each type of sample?

Line 109: "sedate" should be "sedated" and "opiod" should be "opioid"

Line 157-158: What rRNA database was used to assign taxonomic classifications to ASVs? What percentage of ASVs in the dataset were not classified at the genus level? I am concerned that removing ASVs that are unclassified at the genus level is discarding a significant amount of diversity from the dataset, considering that rRNA taxonomy databases are biased towards human-associated bacteria. If the authors have not already done so, I recommend repeating some downstream analyses with the full diversity of ASVs included (i.e., ASV-level beta and alpha diversity across diet regimes, in addition to genus-level), especially if a large percentage of ASVs in the dataset are unclassified at the genus level.

Line 165: Are Bray-Curtis dissimilarities based on an ASV abundance table or a genus-level abundance table?

Line 168: PERMANOVA may be more appropriate because it has a more reliable Type I error rate compared to dbrDA (McArdle and Anderson, 2001). You can control for capture number using the "strata" argument.

Lines 174-176: Did the authors do a correction for multiple comparisons?

Line 186: Please include a bit more detail here so that readers do not need to look up the Flannery & Stagaman methods to follow this section. I recommend including an introductory sentence describing the aim of the CCA analysis and explicitly listing the different types of covariates considered, in addition to referring to table 2.

Lines 202 and 292: The equation has "otu_table" as the dependent variable. Should this actually be "genus table" (if genus level ASV counts were used) or ASV table (if individual ASV level counts were

used)?

Results

Line 212: List the number of samples analyzed and specify the percentage of ASVs that were not classified at the genus level. If a large percentage of ASVs were not classified at the genus level and thus discarded, I recommend repeating downstream analyses (e.g., phylum-level LEfSe analysis) with all ASVs included and including results in the supplement.

Line 241: Should $P > 0.05$ be $P < 0.05$?

Line 255: Does beta-diversity include both intra-individual and inter-individual pairwise sample comparisons?

Line 259: There is not information in the methods concerning how enterotypes were classified. Typically, a clustering analysis (e.g., PAM) is performed on the dissimilarity matrix to determine the optimal number of enterotypes (i.e., clusters). For example, see <https://enterotype.embl.de/enterotypes.html> or Hicks et al. (doi:10.1038/s41467-018-04204-w) "Identification of enterotype" under Statistical Analysis in "Methods" section. Did the authors subjectively decide that there is one enterotype for "restricted nutrition" and one enterotype for "high nutrition"? Given the amount of between-sample variation in the PCoA plot (Fig. 3), there may be multiple enterotypes among the restricted diet microbiome samples.

Line 265: Include the statistical test associated with these p-values.

Line 298: Please include more detail. What was the direction of association for each diet regime and pathogen? Based on Figure 4, microbiome composition differences according to pathogen infection seem more pronounced in the restricted diet regime.

Discussion

See general comments at the beginning of my review.

Line 363: Does sociality (e.g., herd cohesion) change between resource rich and resource deficient time periods?

Figures

General comment: Please include figures of the time series for host covariates and pathogen occurrence/burden over the course of the study (i.e., Fig. S2 expanded to show values for individual capture periods).

Tables

Table 1

Does each sample represent one individual or were multiple samples collected from the same individual during each capture period? Does number of samples refer to fecal samples or all types of samples? If 50-65 buffalo were captured for each sampling period, what was the decision process for which individuals/samples were included in the study? Define NDVI and include information for how NDVI were analyzed in the methods section.

Table 2

Is there a difference between "incidence" and "status"? I recommend specifying that pathogens were dummy coded 1/0 for presence/absence or seroconverted/did not seroconvert (if this is correct). Does "burden" refer to the numbers of parasite worms and/or eggs counted for individual fecal samples?

Supplement

Figure S2: For acute and chronic respiratory pathogens, include the full names in the figure caption.

Reviewers' comments:

Reviewer #1 (Remarks to the Author):

This study examines factors associated with the composition of the gut microbiota in wild African buffalo. Results include the identification of 'enterotype' clusters in the data associated with host diet, as well as associations between gut microbiota composition and respiratory pathogens. It is rare to see a study in which so many interesting data types (microbiome, diet, pathogen status, etc.) are analyzed, and the associations identified will be of broad general interest to the field. The manuscript is well-written and most of the analyses are well explained. However, there is one major issue regarding the identification of enterotypes, as it appears the authors have not tested for the emergence of microbiome clusters from the beta diversity results (revisions/additions suggested below). This issue should be addressed before publication.

In previous analyses, enterotypes have been identified in the data by clustering analyses agnostic to metadata variables. For example, the original Arumugam et al paper use Partitioning Around Medoids (PAM) clustering to identify the enterotypes.

See: <https://enterotype.embl.de/enterotypes.html> for a detailed explanation of the methods and R code. Although the data presented here in the PC plots strongly suggest that the samples do cluster into 2 groups, the PC axes alone (which do not explain all the variation in the underlying beta-diversity measures) are not sufficient to demonstrate this. I would recommend conducting the analyses outlined in the original Enterotype paper (link above). For example, it would be interesting to see what the optimal number of clusters is using Calinski-Harabasz indices. Because the enterotype categories fundamentally rely on the statistical methods employed, it would be best to replicate previous analyses as closely as possible. Then, it would be interesting to test whether clusters that emerge from the data without respect to metadata variables (e.g., diet) are associated with the metadata variables. I think this approach would be a more robust way of addressing the questions 1) do clusters exist in the data and 2) if yes, are the clusters associated with diet/infection status. The analyses already present in the paper could still be presented, as it is also of interest to test for associations between microbiota composition and metadata variables directly.

We have included PAM clustering results and Calinski-Harabasz Index comparisons that demonstrate an optimal number of 2 clusters, consistent with our original manuscript. For the overwhelming majority of samples (85% of the 426 original samples), clusters align by dietary group - i.e. "high nutrition" (either green vegetation or supplemental) vs "restricted nutrition". The methods for this process have been added at lines 189-194 and results at lines 276-279. Our primary interest was in understanding microbiome variation within the context of environmental and dietary variation, therefore we opted to focus our downstream analyses around dietary groups rather than a priori data clusters.

Fecal sampling: Please provide more information on fecal sampling. 8 hrs is a long window for samples to be sitting at ambient conditions before freezing during which time the composition of the microbial community is quickly changing. How did this time vary among samples? Is this variation associated with recovered composition? If this information is not available, this could be discussed as a potential caveat.

We omitted from the original manuscript that samples were placed on ice within 15-30 minutes of collection. They were then frozen at -80 within 8 hours of collection. This detail has been added at line 137.

Figure 1: If available, it would be interesting to include sample locations as points/stars/etc on the map (or as n='some number' for each sampling site), so that the spatial distribution of samples can be readily assessed by readers.

The vast majority of animals were corralled in a corner of the enclosure prior to capture and sampling, therefore sampling location would not be particularly informative for interpreting results of this study. This detail has been included in the caption for figure 1.

Figure 2: It would be interesting to assess taxonomic/genus-level differences between enterotypes as identified by the Arumugam et al. analyses suggested above, in addition to the differences between feed regimes shown here.

As discussed above, enterotypes defined by PAM clustering were 85% aligned with diet, therefore comparing dominant genera between enterotypes did not reveal much additional information relative to what is already shown in figure 2. To informally illustrate this, we identified the top 10 most abundant genera within each of the PAM clusters. We found that 9 out of the top ten genera were shared between cluster 1 and the restricted nutrition group, 8 out of the top ten were shared between cluster 2 and the hay group, and 10/10 between cluster 2 and the green vegetation group.

Figure 3: Similarly, it would be interesting to assess whether individuals shift between enterotypes (as identified by the Arumugam et al. analyses suggested above), and whether these shifts (if they exist) correspond to diet shifts.

Individuals do indeed shift between enterotypes. Repeatability analysis of enterotype membership using the glmm method and logit link in the rptR package yielded a repeatability estimate of 0 ($p = 0.50$), showing that enterotype is not repeatable within individuals. This finding has been added at line 277-278

The intra-individual variation in enterotype associates strongly with diet. The following generalized linear mixed model demonstrated a highly significant relationship between cluster and diet ($p < 2e-16$), after accounting for individual ID (line 279).

glmer(cluster~high nutrition + (1|Animal ID)

Figure 4: It was difficult to assess significance of the difference between positive and negative for each panel. Would it be possible to include stats in the figure and/or legend? As noted for Figures 2 and 3, it would be interesting here to test whether infection status is associated with enterotypes (as identified by the Arumugam et al. analyses suggested above).

We have included envfit p-values on each panel, in addition to further explanation in the legend. Because we were primarily interested in how disease and microbiome composition interact across dietary regimes, we opted not to include the enterotype test suggested above. This decision was made partly for succinctness/clarity, and partly because of the criticisms and concerns about the validity of enterotypes in microbiome literature.

Reviewer #2 (Remarks to the Author):

Couch et al. have analyzed fecal microbiome composition collected over two years in African buffalo fed with supplemental diets. The framework of the study is interesting, the longitudinal data valuable, and the collected disease parameters useful. I don't have major criticisms of the paper, except a few things for the authors to consider (see below). However after reading through this mostly descriptive paper, I feel like I've not really learned anything particularly new about African buffalo gut microbiomes? For example, the study goes into detail showing that there are alpha and beta diversity differences between diet types, which is in itself not terribly surprising, but it doesn't describe clearly in what way they are different besides alpha and beta diversity measures? (see below) And how this is related to seasonal differences besides diet? In addition, I feel like the authors have a highly valuable dataset with collected disease parameters and body condition, but this section of the paper is unfortunately short and does not clearly outline what happens in the gut microbiome in individuals with disease.

We have restructured our narrative to emphasize what we believe are the most impactful findings of this study.

First, the unique circumstances of our study allowed us to separate season from diet and to demonstrate that gut microbiome plasticity associates with dietary fluctuations independent of seasonal environmental variation, which has not been possible in previous studies of the mammalian microbiome. The time frame of our study overlapped an extended period of drought, which resulted in dry season conditions continuing months beyond what is typical for the region. During this time, buffalo were provided with supplemental feed, which resulted in gut microbiomes shifting to the wet season phenotype despite the extremely dry environmental conditions.

Second, our study demonstrates a potential ecological mechanism for the seasonal microbiome plasticity that is observed in wild mammals. While seasonal microbiome changes have been described previously, our large sample size and longitudinal design enabled us to demonstrate that the increase in population-level beta diversity during the dietary restriction maintains population-level gamma diversity despite seasonal loss of individual-level alpha diversity. This finding is significant because it offers an explanation for how microbiome diversity could be maintained over time in social mammals. While individual-level alpha diversity is lost during dietary restriction, individuals can later be recolonized by the microbes that are still present in other members of the population.

Third, as emphasized by this reviewer, our dataset provides valuable information on associations between the microbiome and disease. In addition to the associations we described between microbiome community and disease in our original submission, we have added results an analysis that explicitly links individual microbial taxa with disease.

Major comments

I hope the authors are aware of the controversy surrounding the term Enterotypes. The initial paper describing enterotypes in human gut microbiomes has been extremely criticized, to the point where it is often used as a cautionary example of "how to not analyze microbiome data". I

have personally no problem with using this word, and I believe it is up to the authors to decide how to best describe their results. But thinking how to best help the authors with their revision, it may be worth to at least consider if they still feel like this word is the best way to present the differences they find.

Yes, we are aware of the controversy surrounding this term, however we believe that in light of the common use of the term “Enterotype” within the microbiome research community, it is the most understandable way to describe the compositionally distinct clusters we identified in the dataset. In our analysis, we delineate the microbiome clusters based on diet regime rather than a priori defined enterotypes, therefore we do not believe our conclusions are overly reliant on controversial assumptions about the existence of enterotypes.

How precisely does the microbiome differ with the diet regimes? Which taxa are associated with which diet, and do the results agree with previous diet studies? Why does one of the diet types lead to lower diversity? How does this relate to seasonal changes? The authors have conducted Lefse analyses but do not really present these clearly, except refer to Figure 2a which is not explained particularly well. Is this presented somewhere else and I missed it?

We have attempted to clarify this by further discussion of taxonomic associations with diet at lines 411-427.

A very interesting aspect of this paper is the association tests with diseases. I feel like this section could be stronger, however. What I've gathered from the presented results in figure 4 is that three diseases are associated with the gut microbiome. However, the authors could try to more clearly explain in what way the gut microbiome is different? Are there specific taxa associated with/without disease? What does it mean? Does it vary over the season or between the sexes? What about body condition and disease?

As suggested, we have included results from generalized linear mixed models in table 4, which identified specific genera associated positively and negatively with infection. A discussion of these findings has been included at lines 440-453.

Minor comments

L109: The words “each animal” has been accidentally repeated.

This has been corrected.

L157 & L160: All ASVs not identified to genus level were removed and the rest merged within genera. I'm curious as to the reasoning behind this approach? I have not encountered it previously. Do the authors not miss out on a large proportion of the microbiome data if they require such precise taxonomic assignment? Seeing how the current databases, Silva, GreenGenes, etc are populated with bacteria obtained primarily from model organisms such as humans and mice, a large proportion of unculturable bacteria from wildlife are likely not present with genus-level-specific information. Because of this concern, I'm interested to hear about the reasoning to only keep ASVs with genus information in a wild mammal.

Related to this, a genus-specific approach also complicates the interpretation of the bar plots in Figure 2b. Do these proportions look very different when retaining the excluded genera?

We used this approach because (a) relative abundances of unclassified ASVs were exceedingly low (median relative abundance = 2.34 e-07), therefore the bar plots in figure 2b look very similar (b) initial PAM clustering and diversity analyses were robust to the removal of these ASVs, with 98% of the samples falling into the same cluster regardless of whether ASV or genus-level clustering was used, and (c) removal of unidentified ASVs and merging by genus substantially decreased computational intensity.

L176: PairwiseAdonis package is not cited. In addition, this package seems to be under development still? The developer writes: "This is still a developing version -- results using interactions may not be right. Please validate." Are there any evidence and tests available that this function runs correctly as it says it might not be?

According to the developer's website (<https://github.com/pmartinezarbizu/pairwiseAdonis>), the pairwise.adonis() function is fully functional. It is only pairwise.adonis2() that was still under development at the time of our original submission. We have included a citation at line 209.

There's a large focus in the paper how the microbiome is associated with diet. However, diet differences are here directly associated with seasonal differences. Therefore, I'm curious how much seasonal changes play a role in the gut microbiome in addition to diet? For example there could be a large bacterial fluctuation present in the environment depending on whether it is dry or wet seasons, irrespective of diet. Several previous papers have found large associations between gut microbiome and season in wild mammals.

As explained above, the unique circumstances of our study allowed us to separate season from diet and to demonstrate that gut microbiome plasticity associates with dietary fluctuations independent of seasonal environmental variation, which has not been possible in previous studies of the mammalian microbiome. The time frame of our study overlapped an extended period of drought, which resulted in dry season conditions continuing months beyond what is typical for the region. During this time, buffalo were provided with supplemental feed, which resulted in gut microbiomes shifting to the wet season phenotype despite the extremely dry environmental conditions. As shown in figure 2, the microbiome differences between wet season and feed-supplemented dry season samples were minimal, suggesting that the large associations between the gut microbiome and season are likely driven by dietary availability.

Figure 2: I'm not able to tell the differences between the brown, orange, and yellow colors in the bar plots. Consider using more distinct colors or fewer taxonomic groups in the legend. In addition, why only present 10 randomly selected samples from each feeding regime? I think all host individuals could be useful to present.

We have added all samples from all feeding regimes to the figure. We will change the color palette once we have received the editor's suggestions for best color alternatives for printing.

L367: The authors write that “These results imply that host population size and connectivity may be important for maintaining microbiome diversity within a population”. My question is in what way have the authors analyzed associations with population size? I cannot find this approach in the paper.

While we did not directly assess the effects on population size, a potential connection is implied by our results. We found that the increase in population-level beta diversity during dietary restriction maintains population-level gamma diversity. This suggests that a large, well-connected host population could be important for recolonizing and restoring individual-level alpha diversity following dietary restriction. This explanation has been added at lines 463-469.

Reviewer #3 (Remarks to the Author):

I appreciate the opportunity to read the manuscript submitted to Nature Communications by Couch et al. entitled, ‘Diet drives gut microbiome enterotype shifts at the population level in wild African buffalo’ and offer the following review:

In this study, the authors describe a two-year study, with sampling every 2-3 months, to assess the impact of seasonal fluctuations in diet on gut microbiome composition, pathogen occurrence, and parasite burden in a population of wild African buffalo inhabiting Kruger National Park in South Africa. The sampling frequency (17 capture periods), sampling size (50-65 buffalo per capture period), and numerous types of samples collected and analyzed represent a considerable research effort.

General comments

My main critique is that I’m unsure if this research study, as it’s currently presented, significantly advances our understanding of how dietary shifts influence microbiome composition in wild mammals and/or how pathogen/parasite infections influence or are influenced by microbiome composition. As the authors mention in their manuscript, it’s well established in the microbiome literature that humans and other primates exhibit gut microbiome plasticity in response to seasonality in diet. How does this study contribute novel ecological insight, other than demonstrating seasonality in gut microbiome composition for wild buffalos? Secondly, though the authors found two gut microbial “enterotypes” representative of resource rich and resource poor dietary conditions, it’s unclear if there was a formal statistical analysis to identify enterotypes or if microbiome samples collected during the hay/green vegetation or restricted feed periods were de facto assigned to separate enterotypes.

The reviewer raises two main concerns: (1) lack of clarity regarding the impact and significance of this work, and (2) a need for more robust analysis to identify the existence of host enterotype. We have attempted to address (1) by emphasizing the following novel insights from this study:

First, the unique circumstances of our study allowed us to separate season from diet and to demonstrate that gut microbiome plasticity associates with dietary fluctuations independent of

seasonal environmental variation, which has not been possible in previous studies of the mammalian microbiome. The time frame of our study overlapped an extended period of drought, which resulted in dry season conditions continuing months beyond what is typical for the region. During this time, buffalo were provided with supplemental feed, which resulted in gut microbiomes shifting to the wet season phenotype despite the extremely dry environmental conditions.

Second, our study demonstrates a potential ecological mechanism for the seasonal microbiome plasticity that is observed in wild mammals. While seasonal microbiome changes have been described previously, our large sample size and longitudinal design enabled us to demonstrate that the increase in population-level beta diversity during the dietary restriction maintains population-level gamma diversity despite seasonal loss of individual-level alpha diversity. This finding is significant because it offers an explanation for how microbiome diversity could be maintained over time in social mammals. While individual-level alpha diversity is lost during dietary restriction, individuals can later be recolonized by the microbes that are still present in other members of the population.

Third, as emphasized by this reviewer, our dataset provides valuable information on associations between the microbiome and disease. In addition to the associations we described between microbiome community and disease in our original submission, we have added results an analysis that explicitly links individual microbial taxa with disease.

We have addressed (2) by including results of a formal clustering analysis based on Arumugam et al (2011), which robustly identifies the presence of two enterotypes that associate very strongly with enriched vs restricted dietary availability.

Abstract

General comments: I recommend including more details about the study population (i.e., X individuals were longitudinally sampled from a long-term study population in Kruger National Park), sampling regime (e.g., 17 captures over 2 years), and the types of samples collected. For example, though the authors mention respiratory pathogens at the end of the abstract, the abstract does not explicitly mention that multiple types of samples were collected, in addition to fecal samples. Can you explicitly mention that resource rich and resource restricted periods associated with wet and dry seasons, respectively?

Line 5: I recommend rewording as “changes in dietary resources”

This change has been made

Lines 8 and 10: It's unclear if beta diversity refers to both intra-host, inter-host, or both inter- and intra-host microbial dissimilarities.

This has been changed to “between-host and within-host beta diversity”

Lines 13-16: Was pathogen detection associated with resource rich periods or restricted dietary conditions? Could respiratory pathogen detection be confounded by environmental conditions and not necessarily be associated with gut microbiome composition?

We included enriched vs restricted dietary period as a conditional variable in our CCA model, thus controlling for any association between pathogen detection and resource conditions.

Introduction

Line 71: I recommend including much more detail about the buffalo study population and sampling regime.

More detail has been added (lines 68-81)

Line 84: It's unclear to me how respiratory pathogen infection would be directly influenced by changes in gut microbiome composition.

This has been clarified at lines 96-98: "Changes to microbiome-pathogen relationships could be mediated by resource-driven variation in the immune system, similar to environmentally-driven changes observed in predator-prey dynamics (Bastille-Rousseau et al. 2017). Additionally, changes in host diet could alter competition for resources between commensal and pathogenic microbes, or enable facilitative interactions (DuBow 1988)."

Methods

Line 96: Include the total number of capture periods here (according to Table 1, there are 17). How many individual animals were longitudinally sampled, for each type of sample?

Information on the number of captures and number of animals sampled are now included on line 107-111.

Line 109: "sedate" should be "sedated" and "opiod" should be "opioid"

This has been corrected.

Line 157-158: What rRNA database was used to assign taxonomic classifications to ASVs? What percentage of ASVs in the dataset were not classified at the genus level? I am concerned that removing ASVs that are unclassified at the genus level is discarding a significant amount of diversity from the dataset, considering that rRNA taxonomy databases are biased towards human-associated bacteria. If the authors have not already done so, I recommend repeating some downstream analyses with the full diversity of ASVs included (i.e., ASV-level beta and alpha diversity across diet regimes, in addition to genus-level), especially if a large percentage of ASVs in the dataset are unclassified at the genus level.

Silva v 132 was used for classification. We opted to exclude unclassified genera because (a) relative abundances of unclassified ASVs were exceedingly low (median relative abundance = 2.34×10^{-7}), (b) initial clustering and diversity analyses were robust to the removal of these ASVs, with 98% of the samples falling into the same cluster regardless of whether ASV or genus-level clustering was used, and (c) removal of unidentified ASVs and merging by genus substantially decreased computational intensity.

Line 165: Are Bray-Curtis dissimilarities based on an ASV abundance table or a genus-level abundance table?

Genus-level abundance table. This information has been added at line 243 and 362. However, as described above, dissimilarities were robust to the use of an ASV abundance table and yielded similar beta diversity and clustering results.

Line 168: PERMANOVA may be more appropriate because it has a more reliable Type I error rate compared to dbRDA (McArdle and Anderson, 2001). You can control for capture number using the "strata" argument.

This is an intelligent suggestion, however, we opted to use dbRDA rather than PERMANOVA because the latter is sensitive to differences in beta dispersion. Permutation tests demonstrated significant differences in beta dispersion between diet regimes, thus undermining our PERMANOVA results.

Lines 174-176: Did the authors do a correction for multiple comparisons?

Yes, we have added this information at line 210.

Line 186: Please include a bit more detail here so that readers do not need to look up the Flannery & Stagaman methods to follow this section. I recommend including an introductory sentence describing the aim of the CCA analysis and explicitly listing the different types of covariates considered, in addition to referring to table 2.

More detail has been provided at lines 222-228.

Lines 202 and 292: The equation has "otu_table" as the dependent variable. Should this actually be "genus table" (if genus level ASV counts were used) or ASV table (if individual ASV level counts were used)?

Yes, we have made the suggested change.

Results

Line 212: List the number of samples analyzed and specify the percentage of ASVs that were not classified at the genus level. If a large percentage of ASVs were not classified at the genus level and thus discarded, I recommend repeating downstream analyses (e.g., phylum-level LEfSe analysis) with all ASVs included and including results in the supplement.

Initial PAM clustering results were robust to the removal of these ASVs, with 98% of the samples falling into the same cluster regardless of whether all ASVs or known genera were used. Phylum-level LEfSe analysis of all ASVs classified to the phylum level identified 8/9 differentially abundant phyla among known genera. This information has been included at lines 273-274 and 297-301.

Line 241: Should $P > 0.05$ be $P < 0.05$?

Yes, this has been corrected

Line 255: Does beta-diversity include both intra-individual and inter-individual pairwise sample comparisons?

Yes, this has been clarified.

Line 259: There is not information in the methods concerning how enterotypes were classified. Typically, a clustering analysis (e.g., PAM) is performed on the dissimilarity matrix to determine the optimal number of enterotypes (i.e., clusters). For example, see <https://enterotype.embl.de/enterotypes.html> or Hicks et al. (doi:10.1038/s41467-018-04204-w) “Identification of enterotype” under Statistical Analysis in “Methods” section. Did the authors subjectively decide that there is one enterotype for “restricted nutrition” and one enterotype for “high nutrition”? Given the amount of between-sample variation in the PCoA plot (Fig. 3), there may be multiple enterotypes among the restricted diet microbiome samples.

We compared Calinski-Harabasz index values for 2 – 20 clusters, which identified an optimum number of 2 clusters in both the ASV-level and genus-level datasets (lines 276-277).

Line 265: Include the statistical test associated with these p-values.

These results are based on permutation tests for homogeneity of multivariate dispersions. This information has been added at line 324-325.

Line 298: Please include more detail. What was the direction of association for each diet regime and pathogen? Based on Figure 4, microbiome composition differences according to pathogen infection seem more pronounced in the restricted diet regime.

Directionality is difficult to interpret in CCA results, therefore we included results from generalized linear mixed models assessing relationships between each bacterial genera and each of the three diseases of interest. These results are included in table 4 and at lines 370-376.

Discussion

See general comments at the beginning of my review.

The discussion has been revised substantially to emphasize the novel contributions of our study, as described above.

Line 363: Does sociality (e.g., herd cohesion) change between resource rich and resource deficient time periods?

Social dynamics likely change between feed regimes, however, we did not explicitly measure social interactions in this study.

Figures

General comment: Please include figures of the time series for host covariates and pathogen occurrence/burden over the course of the study (i.e., Fig. S2 expanded to show values for individual capture periods).

The requested figures (S3 and S4) have been added to the supplementary materials.

Tables

Table 1

Does each sample represent one individual or were multiple samples collected from the same individual during each capture period?

We have clarified in the legend: "Each individual was sampled only once at each capture period."

Does number of samples refer to fecal samples or all types of samples?

We have clarified in the column heading: "number of fecal samples".

If 50-65 buffalo were captured for each sampling period, what was the decision process for which individuals/samples were included in the study?

We have clarified in the methods and in the figure legend that all microbiome samples were included in alpha and beta diversity analyses and envfit analysis, but that only samples with complete covariate datasets were used in the CCA.

Define NDVI and include information for how NDVI were analyzed in the methods section.

NDVI has been defined in the table legend, and analysis has been included at lines 121-127 in the methods section.

Table 2

Is there a difference between "incidence" and "status"? I recommend specifying that pathogens were dummy coded 1/0 for presence/absence or seroconverted/did not seroconvert (if this is correct).

Yes, pathogens are recorded as either presence/absence (chronic) or seroconversion status (acute). This information has been added to the table legend.

Does "burden" refer to the numbers of parasite worms and/or eggs counted for individual fecal samples?

Yes, burden refers to eggs per gram in individual fecal samples. This information has been added to the table legend.

Figure S2: For acute and chronic respiratory pathogens, include the full names in the figure caption.

This has been added.

REVIEWER COMMENTS

Reviewer #1 (Remarks to the Author):

The authors have addressed all of my comments from the previous round of review.

Reviewer #2 (Remarks to the Author):

I'm happy with the changes the authors made to the revised manuscript and I think they have adequately addressed the questions and concerns raised. I only have a few minor comments which seem to be simple errors the authors made during revision.

Minor comments:

- L127: R packages seem to have been intended to cite here, but then accidentally forgotten.
- L286: Is the sentence about PAM clustering and correlation with dietary regime in the correct result paragraph? Or should it be in the following paragraph titled "Dietary regime is the primary driver of microbiome structure in African buffalo:"?
- L430 talks about the genus Ruminococcaceae UCG-005 and L443 discusses Ruminococcus UCG-005. Is this a mistake?
- Other: The information given within the Reporting Summary form does not match the information given in the manuscript, with regards to some details like study period, analyses. Please update the Reporting Summary form.

Reviewer #3 (Remarks to the Author):

I appreciate the authors' careful attention in addressing my and the other reviewers' comments on their first submission. However, I have one lingering concern about their methodology:

Major comments

Lines 222-257: Perhaps I'm fundamentally misunderstanding the authors' approach here but it seems to include two common statistical mistakes: 1) interpreting comparisons between two effects without directly comparing them (samples from different dietary regimes are tested separately), and 2) circular analysis ("GLMMs were run only on samples from the dietary regime in which the envfit analysis identified significant correlation with the disease of interest"). See <https://doi.org/10.7554/eLife.48175.005>. Thus, the reported associations between certain bacteria genera and pathogens may be statistical artefacts. With the caveat that I have not encountered this particular methodology before, I consider it appropriate to only report statistical associations from the pooled datasets and to explore all possible associations between each pathogen and each bacteria

genera, while correcting for multiple testing.

Though the authors state that they found qualitatively similar downstream results when including the full diversity of ASVs versus filtering out unclassified genera, I recommend that they include results/figures pertaining to the full diversity ASVs (e.g., PAM clustering of enterotypes) in the supplement.

Minor comments:

Line 14: I recommend rewording as “may impact” instead of “impact”, given that the mechanism is unclear concerning why three specific pathogens are associated with diet regime.

Line 270: Were these ~30,000 ASVs shared among samples or primarily singletons?

Line 279: I recommend rewording as “demonstrated a significant relationship between cluster membership and diet regime at the population level.”

Discussion: Though the authors responded to my previous review that they did not explicitly measure buffalo social dynamics, I recommend including a speculative comment about herd cohesion differing between dietary regimes and citing any relevant sources, if they exist.

REVIEWER COMMENTS

Reviewer #1 (Remarks to the Author):

The authors have addressed all of my comments from the previous round of review.

Reviewer #2 (Remarks to the Author):

I'm happy with the changes the authors made to the revised manuscript and I think they have adequately addressed the questions and concerns raised. I only have a few minor comments which seem to be simple errors the authors made during revision.

Minor comments:

- L127: R packages seem to have been intended to cite here, but then accidentally forgotten.

The missing references have been added at lines 120-121

- L286: Is the sentence about PAM clustering and correlation with dietary regime in the correct result paragraph? Or should it be in the following paragraph titled “Dietary regime is the primary driver of microbiome structure in African buffalo:”?

We have moved the sentence about PAM clustering to the following paragraph. It now appears at lines 299-305

- L430 talks about the genus Ruminococcaceae UCG-005 and L443 discusses Ruminococcus UCG-005. Is this a mistake?

There was a mistake at line 443 – the sentence at L443 has been corrected to read Ruminococcaceae UCG-005 (now appears at line 453)

- Other: The information given within the Reporting Summary form does not match the information given in the manuscript, with regards to some details like study period, analyses. Please update the Reporting Summary form.

The Reporting Summary form has been updated.

Reviewer #3 (Remarks to the Author):

I appreciate the authors' careful attention in addressing my and the other reviewers' comments on their first submission. However, I have one lingering concern about their methodology:

Major comments

Lines 222-257: Perhaps I'm fundamentally misunderstanding the authors' approach here but it seems to include two common statistical mistakes: 1) interpreting comparisons between two effects without directly comparing them (samples from different dietary regimes are tested separately), and 2) circular analysis ("GLMMs were run only on samples from the dietary regime in which the envfit analysis identified significant correlation with the disease of interest"). See <https://doi.org/10.7554/eLife.48175.005>. Thus, the reported associations between certain bacteria genera and pathogens may be statistical artefacts. With the caveat that I have not encountered this particular methodology before, I consider it appropriate to only report statistical associations from the pooled datasets and to explore all possible associations between each pathogen and each bacteria genera, while correcting for multiple testing.

The reviewer's response suggests a need to clarify the purpose of the envfit and GLMM analyses in the methods section, as we believe their concerns stem from miscommunication on our part. We have revised our methods section to address the reviewer's concerns 1&2 as described below:

- 1) *Interpreting comparisons between two effects without directly comparing them (samples from different dietary regimes are tested separately):*

The reviewer raises a valid concern which we believe can be addressed by clarifying our methodology in the manuscript. We did not rely on the envfit analysis to statistically compare how effects differed among diet regimes, but rather to inform data reduction before parameterizing a CCA model to statistically compare effects. Due to our initially large covariate dataset, it was necessary to reduce the number of covariates in our dataset to minimize the risk of model overfitting or identification of spurious correlations. Because of the strikingly different community composition of high versus low nutrition regimes demonstrated by unconstrained ordination (PCA) visualization and clustering analysis, we expected to find different patterns of covariate correlations in each dietary regime. Therefore, to avoid obscuring any patterns that were present under only a single diet regime, we ran the exploratory envfit analyses on each diet regime separately, in addition to the pooled dataset. Upon re-examining the manuscript, we recognize that it was unclear that envfit was run on the pooled dataset in addition to the separate diet regimes, and that the purpose of the envfit was to inform data reduction rather than to explicitly compare covariate correlations. We have clarified our approach in the manuscript at lines 222-244.

- 2) *Circular analysis ("GLMMs were run only on samples from the dietary regime in which the envfit analysis identified significant correlation with the disease of interest"):*

As with first concern, we believe the second concern raised by this reviewer can be addressed by clarifying our reasons for conducting the GLMM analysis and the conclusions we drew from them. The intent of the GLMM analysis was not to further

support the hypothesis that overall microbiome composition associates with the three pathogens in question, but rather to identify the bacterial genera potentially driving the patterns detected in the CCA analysis. In addition to addressing different questions, the ordination-based analyses (including envfit) had different response variables (overall community composition) than did the GLMMs (specific bacterial taxa), therefore circularity should not be a concern in this case. Our rationale and approach have been clarified at lines 257-260 of the manuscript.

Though the authors state that they found qualitatively similar downstream results when including the full diversity of ASVs versus filtering out unclassified genera, I recommend that they include results/figures pertaining to the full diversity ASVs (e.g., PAM clustering of enterotypes) in the supplement.

We have included ASV-level PAM clustering results as suggested by the reviewer. However, upon consideration, this information seemed to fit better in the main text rather in the supplement, therefore it has been included at lines 303-305 rather than in the supplement.

Minor comments:

Line 14: I recommend rewording as “may impact” instead of “impact”, given that the mechanism is unclear concerning why three specific pathogens are associated with diet regime.

The wording has been updated to “may impact” at line 14 per the reviewer’s recommendation.

Line 270: Were these ~30,000 ASVs shared among samples or primarily singletons?

21% of unclassified ASVs were singletons, and 91% occurred less than ten times in the entire dataset. This information has been added at lines 283-285.

Line 279: I recommend rewording as “demonstrated a significant relationship between cluster membership and diet regime at the population level.”

The suggested wording change has been made at lines 301-302.

Discussion: Though the authors responded to my previous review that they did not explicitly measure buffalo social dynamics, I recommend including a speculative comment about herd cohesion differing between dietary regimes and citing any relevant sources, if they exist.

The following comment has been added at lines 441-446: “We did not explicitly measure behavioral changes in this study, but free-ranging buffalo exhibit seasonal changes in behavior and herd cohesion (Turner et al 2005). It is possible that if such behavioral patterns are present in our study herd, they could mediate changes in microbial transmission dynamics, potentially

explaining some of the diet-associated patterns in richness and composition found in this study (Tung et al. 2015, Moeller et al. 2016).”

REVIEWERS' COMMENTS

Reviewer #3 (Remarks to the Author):

I appreciate the clarification concerning the authors' methodology. They have addressed all of my comments from the previous review.